# Definition of two agonist types at the mammalian cold-activated channel TRPM8

Annelies Janssens[1†], Maarten Gees[1†], Balazs Istvan Toth[1†], Debapriya Ghosh[1], Marie Mulier[1], Rudi Vennekens[1], Joris Vriens[1,2], Karel Talavera[1], Thomas Voets[1*]

[1]Laboratory of Ion Channel Research and TRP channel Research Platform Leuven, Department of Cellular and Molecular Medicine, University of Leuven, Leuven, Belgium; [2]Laboratory of Experimental Gynaecology, University of Leuven, Leuven, Belgium

**Abstract** Various TRP channels act as polymodal sensors of thermal and chemical stimuli, but the mechanisms whereby chemical ligands impact on TRP channel gating are poorly understood. Here we show that AITC (allyl isothiocyanate; mustard oil) and menthol represent two distinct types of ligands at the mammalian cold sensor TRPM8. Kinetic analysis of channel gating revealed that AITC acts by destabilizing the closed channel, whereas menthol stabilizes the open channel, relative to the transition state. Based on these differences, we classify agonists as either type I (menthol-like) or type II (AITC-like), and provide a kinetic model that faithfully reproduces their differential effects. We further demonstrate that type I and type II agonists have a distinct impact on TRPM8 currents and TRPM8-mediated calcium signals in excitable cells. These findings provide a theoretical framework for understanding the differential actions of TRP channel ligands, with important ramifications for TRP channel structure-function analysis and pharmacology.

*For correspondence: thomas. voets@med.kuleuven.be

†These authors contributed equally to this work

Competing interests: The authors declare that no competing interests exist.

## Introduction

Neurons of the somatosensory system act as individually tuned sensory cells that convert specific thermal, mechanical and/or chemical stimuli into electrical signals, which are then conveyed to the central nervous system (*Vriens et al., 2014*). Within the somatosensory system, several members of the TRP superfamily of cation channels act as polymodal molecular sensors of both temperature, and a variety of endogenous and exogenous chemicals, including a plethora of plant-derived compounds (*Clapham, 2003*; *Tominaga et al., 1998*; *Voets et al., 2005*; *Vriens et al., 2014*). Chemical activation of TRP channels in nerve endings of trigeminal or dorsal root ganglion neurons is generally believed to underlie typical chemesthetic sensations evoked by such plant-derived substances (*Bandell et al., 2007*), such as the burning heat evoked by capsaicin (the pungent substance in hot peppers), which acts as a selective agonist of the heat-activated TRPV1 (*Caterina et al., 1997*), and the cool sensation evoked by menthol (the cooling compound in mint plants), due to activation of the cold sensor TRPM8 (*McKemy et al., 2002*; *Peier et al., 2002*). Such TRP channel ligands are present in widely used foodstuffs and drugs (*Nilius and Appendino, 2013*), and are extensively used as pharmacological tools to study somatosensation and/or TRP channel function in vitro and in vivo (*Julius, 2013*). Yet, very little is known about the molecular and biophysical mechanisms of action of the various TRP channel ligands.

We studied the agonist effects of AITC, also known as mustard oil, a pungent organosulphur compound derived from *Brassica* plants. AITC is responsible for the characteristic oral sensations that one experiences upon eating Dijon mustard or wasabi, which contain between 5–30 mM of AITC (*Uematsu et al., 2002*). Whereas earlier work has firmly established that AITC activates TRPA1 and TRPV1 in nociceptor neurons, approximately 10% of dorsal root ganglion neurons remained

**eLife digest** Sensory neurons in our skin detect cues from the environment – such as temperature and touch – and pass the information onto other cells in the nervous system. A protein called TRPM8 in sensory neurons is responsible for our ability to detect cool temperatures. TRPM8 sits in the membrane that surrounds the cell and forms a channel that can allow sodium and calcium ions to enter the cell. Cold temperatures activate TRPM8, which opens the channel and triggers electrical activity in the sensory neurons.

Chemicals that cause a cold sensation – such as menthol, the refreshing substance found in mint plants – can also open the TRPM8 channel. Janssens, Gees, Toth et al. investigated how menthol, and another natural compound called mustard oil, influence the opening of TRPM8. The experiments show that menthol and mustard oil both stimulate sensory neurons by opening the TRPM8 ion channel, but using different mechanisms. Mustard oil forces the channel to open faster than it normally would, whereas menthol prevents the channel from closing. Further experiments show that these mechanisms explain why some compounds stimulate sensory neurons more strongly than others.

The findings of Janssens, Gees, Toth et al. will help to understand how chemicals act on this class of ion channels, and how this affects the roles of the ion channels in cells. Altering the activity of TRPM8 and related ion channels may help to reduce pain in humans so a future challenge is to use these new insights to develop drugs that target these channels more efficiently.

AITC-responsive after combined genetic deletion these two TRP channels (*Bandell et al., 2004*; *Bautista et al., 2006*; *Everaerts et al., 2011*; *Jordt et al., 2004*). In this work we show that AITC excites this subset of somatosensory neurons via direct activation of TRPM8. Interestingly, a detailed biophysical analysis revealed that AITC activates TRPM8 by inducing a relative destabilization of the closed conformation relative to the transition state. This mode of action is fundamentally different from that of other known TRPM8 agonists such as menthol, which stabilize the open conformation relative to the transition state. Based on these results, we propose to classify TRPM8 agonists as either type I (menthol-like) or type II (AITC-like), and provide a kinetic model that accurately describes the differential actions of the two agonist types on channel gating kinetics. Finally, we illustrate that the two agonist types have a distinct impact on TRPM8-mediated currents and calcium signals in excitable cells.

## Results

### TRPM8-dependent responses to AITC in sensory neurons

To investigate the origin of TRPV1- and TRPA1-independent AITC responses, we performed $Ca^{2+}$ imaging experiments on dorsal root ganglion (DRG) neurons isolated from TRPV1/TRPA1 double knockout mice. In line with previous work (*Everaerts et al., 2011*), we found that a small fraction of these TRPV1/TRPA1-deficient neurons (55 out of 578; 9% ) showed a rapid and reversible increase in intracellular $Ca^{2+}$ in response to 3 mM AITC (*Figure 1A*). These AITC-responsive cells consistently responded to menthol (54 out of 55; 98%) (*Figure 1A,B*). In these cells, the responses to both AITC and menthol were fully inhibited by the TRPM8 antagonist AMTB, and recovered partially upon AMTB washout (*Figure 1C*). Taken together, these results indicate that TRPV1- and TRPA1-independent AITC responses in DRG neurons depend on the cold- and menthol-sensitive channel TRPM8.

### AITC activates heterologously expressed TRPM8

To investigate the mechanisms underlying TRPM8-dependent AITC sensitivity in sensory neurons, we tested the effect of acute application of AITC on whole-cell currents in HEK293 cells heterologously expressing human TRPM8. At room temperature, TRPM8 exhibits substantial activity, which can be recorded as an outwardly rectifying current (*Figure 2A,B*). Application of AITC at concentrations $\geq$300 μM caused a rapid and reversible increase in TRPM8 current (*Figure 2A*). The amplitude of the response increased with AITC concentration, with relatively stronger effects at negative

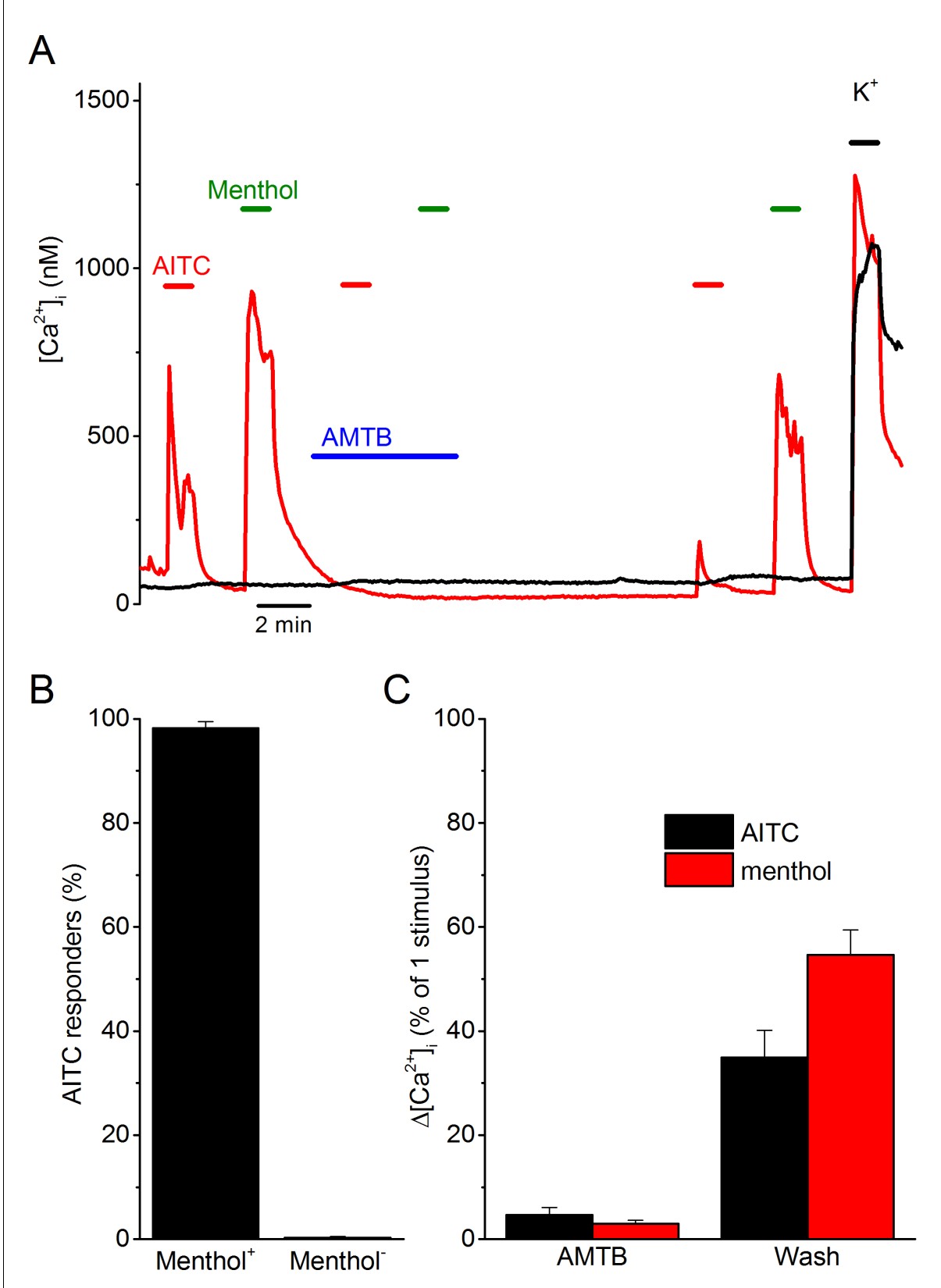

**Figure 1.** AITC excites trigeminal neurons in a TRPM8-dependent manner. (**A**) Examples of fura-2-based intracellular calcium measurements in trigeminal neurons from TRPV1/TRPA1 double knockout mice. The red trace represents a neuron that shows responses to AITC (3 mM) and menthol (50

*Figure 1 continued on next page*

*Figure 1 continued*

μM), which can be reversible inhibited by AMTB (2 μM). The black trace represents a non-responder. A high K⁺-solution (50 mM K⁺) was used at the end of the experiments to identify neurons from non-neuronal cells. In total, 578 neurons from 6 different mice were analyzed. (**B**) Percentage of AITC-responsive neurons in menthol-sensitive (n = 55) and menthol-insensitive (n = 523) neurons. (**C**) Quantification of the reversible inhibition by AMTB of responses to AITC and menthol (n = 54).

voltages, but did not saturate at the highest concentration tested (10 mM; *Figure 2C*). At 3 and 10 mM AITC, activation was followed by a gradual decay of TRPM8 current, reducing current amplitude to levels below the basal level (*Figure 2A,B*). Following washout of 3 or 10 mM AITC after prolonged exposure, we observed a rapid initial decrease in current followed by a gradual restoration of the current to the basal level (*Figure 2A*), suggesting that the agonistic effect of AITC reverses more rapidly than the inhibitory effect. Rapid and reversible current responses to AITC were also observed in cell-free inside-out patches from human TRPM8-expressing HEK293 cells, indicating that the effect of AITC on TRPM8 is membrane-delimited (*Figure 2D,E*).

It has been put forward that AITC induces trafficking of TRPA1 to the plasma membrane (*Schmidt et al., 2009*). To test whether AITC-induced activation of TRPM8 also involves rapid translocation of the channel towards the plasma membrane, we expressed human TRPM8 coupled with mCherry at its C terminus (TRPM8-mCherry), and performed total internal reflection fluorescence (TIRF) microscopy to monitor potential AITC-induced transport of TRPM8 towards the plasma membrane. We have recently shown that TRPM8-mCherry is fully functional, and can be used to track cellular TRPM8 transport (*Ghosh et al., 2016*). As shown in *Figure 2F,G*, application of 3 mM AITC had no detectable effect on the TRPM8-mCherry fluorescence in the close vicinity of the plasma membrane. mCherry fluorescence amounted to 99 ± 1% and 102 ± 2% of the pre-AITC level after 5 and 50 s of AITC application, respectively. Since the onset of current activation by AITC was very rapid, with maximal current achieved within ~2 s (*Figure 2A,D*), we can exclude a significant contribution of trafficking to the acute agonistic effect of AITC on TRPM8.

## Distinct effects of AITC and menthol on gating kinetics

To investigate the mechanism of the agonistic effect of AITC in more detail, we recorded TRPM8 currents during voltage steps ranging from −140 to +220 mV, both in control conditions and immediately upon application of AITC (*Figure 3A*). Analysis of the steady-state conductances revealed that AITC has little or no effect on the maximal conductance at strongly depolarizing potentials, but shifts the voltage-dependent activation curves towards more negative voltages in a concentration-dependent manner (*Figure 3B,C*). Such ligand-induced shifts in the voltage-dependent activation curve have been shown earlier to describe the effects of agonists on TRP channels, including the effect of menthol on TRPM8 (*Voets et al., 2004*; *Voets et al., 2007*) (*Janssens and Voets, 2011*).

However, when analyzing the kinetics of TRPM8 current activation/deactivation during voltage steps in more detail, we observed a remarkable difference between the effects of menthol and AITC. This is illustrated in *Figure 4A*, which provides a comparison of currents in the absence of ligands and in the presence of either 3 mM AITC or 30 μM menthol, concentrations that provoke similar steady-state TRPM8 current amplitudes at the end of the voltage steps. In the presence of AITC we observed a clear acceleration of the gating kinetics upon depolarization to +120 mV, whereas the current relaxation kinetics upon repolarization to −80 mV were not markedly altered. In stark contrast, in the presence of menthol we found a pronounced slowing of the kinetics of current relaxation, most noticeable upon repolarization to −80 mV (*Figure 4A,B*).

To quantify the differences in gating kinetics in more detail, we fitted exponential functions to the current time courses during voltage steps. In line with earlier work, we found that in the absence of ligands the time courses at +120 and −80 mV were generally well fitted by a mono-exponential function (*Figure 4A,B*; *Figure 4—figure supplement 1*), yielding the time constants of current relaxation at both potentials ($\tau_{+120\ mV}$ and $\tau_{-80\ mV}$). In the presence of 3 mM AITC, the time courses remained well fitted by a mono-exponential function, and the accelerated kinetics were reflected in a reduction of $\tau_{+120\ mV}$ compared to control, whereas $\tau_{-80\ mV}$ was unaltered (*Figure 4A,B,E*). In contrast, the relaxation kinetics in the presence of 30 μM menthol were consistently slower than in control and were no longer mono-exponential: at least two exponential terms were required to accurately

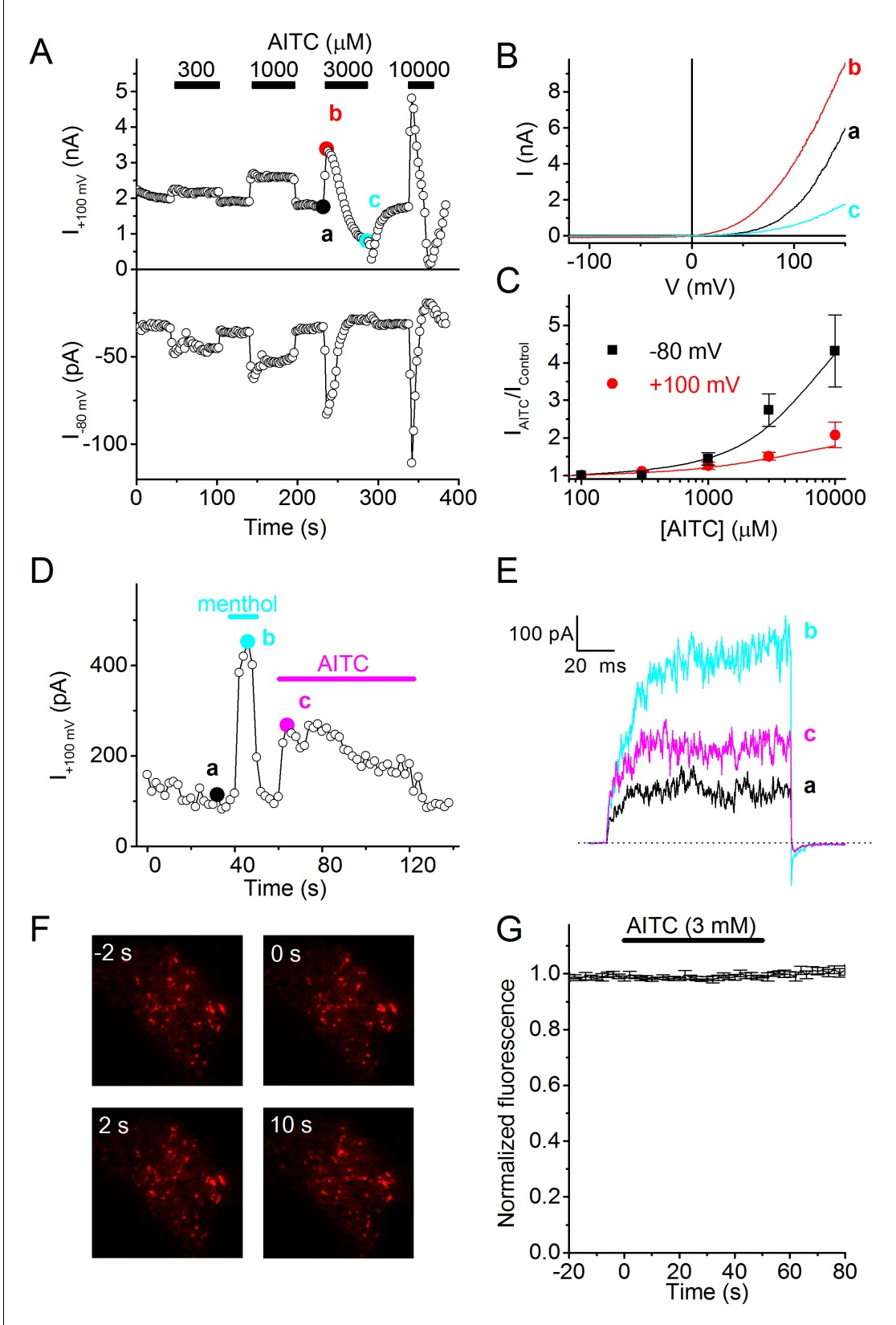

**Figure 2.** AITC activates human TRPM8. (A) Time course of whole-cell currents at +100 and −80 mV in HEK293 cells expressing human TRPM8, upon stimulation with the indicated concentrations of AITC. (B) Current-voltage relations recorded at the time points indicated in (A). (C) Relative AITC-

*Figure 2 continued on next page*

*Figure 2 continued*

induced current increase at +100 and −80 mV (n = 9). (**D**) Menthol (50 μM) and AITC (3 mM) activate TRPM8 in cell-free inside-out patches during repetitive 100-ms voltage steps to +100 mV. Comparable current activation was measured in 5 out of 5 inside-out patches. (**E**) Current traces recorded at the time points indicated in (**D**). (**F**) TIRF images showing mCherry-tagged human TRPM8 in the perimembrane region before and during stimulation with 3 mM AITC. Micrographs are 20 × 20 μm. (**G**) Lack of change in perimembrane mCherry-fluorescence during stimulation with AITC (n = 6). Fluorescence was normalized to the total fluorescence before adding AITC to the bath solution.

describe the current time course at +120 ($\tau_{+120\ mV,fast}$ and $\tau_{+120\ mV,slow}$) and −80 mV ($\tau_{-80\ mV,fast}$ and $\tau_{-80\ mV,slow}$) (*Figure 4A,B,F*). These distinct effects of AITC and menthol on the current relaxation kinetics of TRPM8 were observed over a broad concentration range (*Figure 4C–F*). Other known TRPM8 agonists, including thymol, icilin, and linalool, act in a similar manner as menthol, slowing down the kinetics of activation and deactivation, albeit less pronounced (*Figure 4—figure supplement 2*).

## Defining Type I and Type II agonists

Based on these results, we propose that TRPM8 agonists be classified into two types based on their effect on the gating kinetics: Type I (menthol-like) agonists induce a slowing of the gating kinetics, which is most prominently observed as slowly deactivating tail currents following repolarization, whereas Type II (AITC-like) agonists cause an acceleration of the kinetics of channel activation upon depolarization, with little or no effect on the kinetics of deactivating tail currents (*Figure 4*). The differential effects of the two ligand types on the gating kinetics suggest that they act on different conformational states of the channel during the gating process. In particular, the characteristic slowly decaying tail currents upon repolarization in the presence of menthol indicate that menthol impedes voltage-dependent channel deactivation, which points at a stabilization of the channel in an open conformation. Oppositely, the faster current relaxation upon depolarization in the presence of AITC indicates that AITC accelerates voltage-dependent channel activation, which points at a destabilization of the channel in a closed conformation.

To further pinpoint the mechanistic basis of different effects of Type I and Type II agonists on channel gating kinetics, we built on a previously described voltage-dependent Monod-Wyman-Changeux (MWC) model that was initially developed to describe the concerted actions of $Ca^{2+}$ and voltage on the gating of large conductance $Ca^{2+}$-activated potassium (BK) channels (*Cox et al., 1997*). We have shown earlier that this model can accurately describe the effect of menthol, voltage and temperature on steady-state TRPM8 currents (*Janssens and Voets, 2011*). Moreover, based on analysis of channel chimeras with different combinations of wild type and mutated menthol binding sites, it was found that a single TRPM8 channel can bind up to four ligand molecules, each subunit having and ligand binding site with an affinity $K_d$ (in the closed state), and that every bound ligand shifts the equilibrium between the closed and open channel by a similar extent (*Janssens and Voets, 2011*). The energetic effect of ligand binding can be quantified as $\Delta\Delta G_{ligand}$, which represents the change of the difference in Gibbs free energy between the closed and open state of the channel ($\Delta G$) upon binding of one ligand molecule to one of the four subunits. In the case of an agonist, $\Delta\Delta G_{ligand} < 0$, which implies that the open state becomes more stable relative to the closed state. As illustrated by the energy diagrams in *Figure 5A*, a negative $\Delta\Delta G_{ligand}$ can be the result of a ligand-induced relative stabilization of the open state, destabilization of the closed state, or a combination of both, taking the transition state as the reference. In the case of a relative stabilization of the open state, the energy barrier for the transition from open to closed will become higher, which would lead to slower closing rates, as seen with the Type I agonists (*Figure 5A*). Oppositely, relative destabilization of the closed state will reduce the energy barrier for the transition from closed to open, which would be reflected in faster opening rates, as seen with the Type II agonist AITC (*Figure 5—figure supplement 1*).

We performed global fits of the MWC model to the experimental current time courses obtained in individual cells during voltage steps in both the absence and presence of different concentrations of AITC or menthol. Values for $K_{d,menthol}$, $\Delta\Delta G_{menthol}$, $K_{d,AITC}$ and $\Delta\Delta G_{AITC}$ were obtained from shifts in the steady-state voltage-dependent activation curves (*Janssens and Voets, 2011*). Note that values for the on rates for ligand binding ($k_{on}$) were determined from the fits, in contrast to earlier work

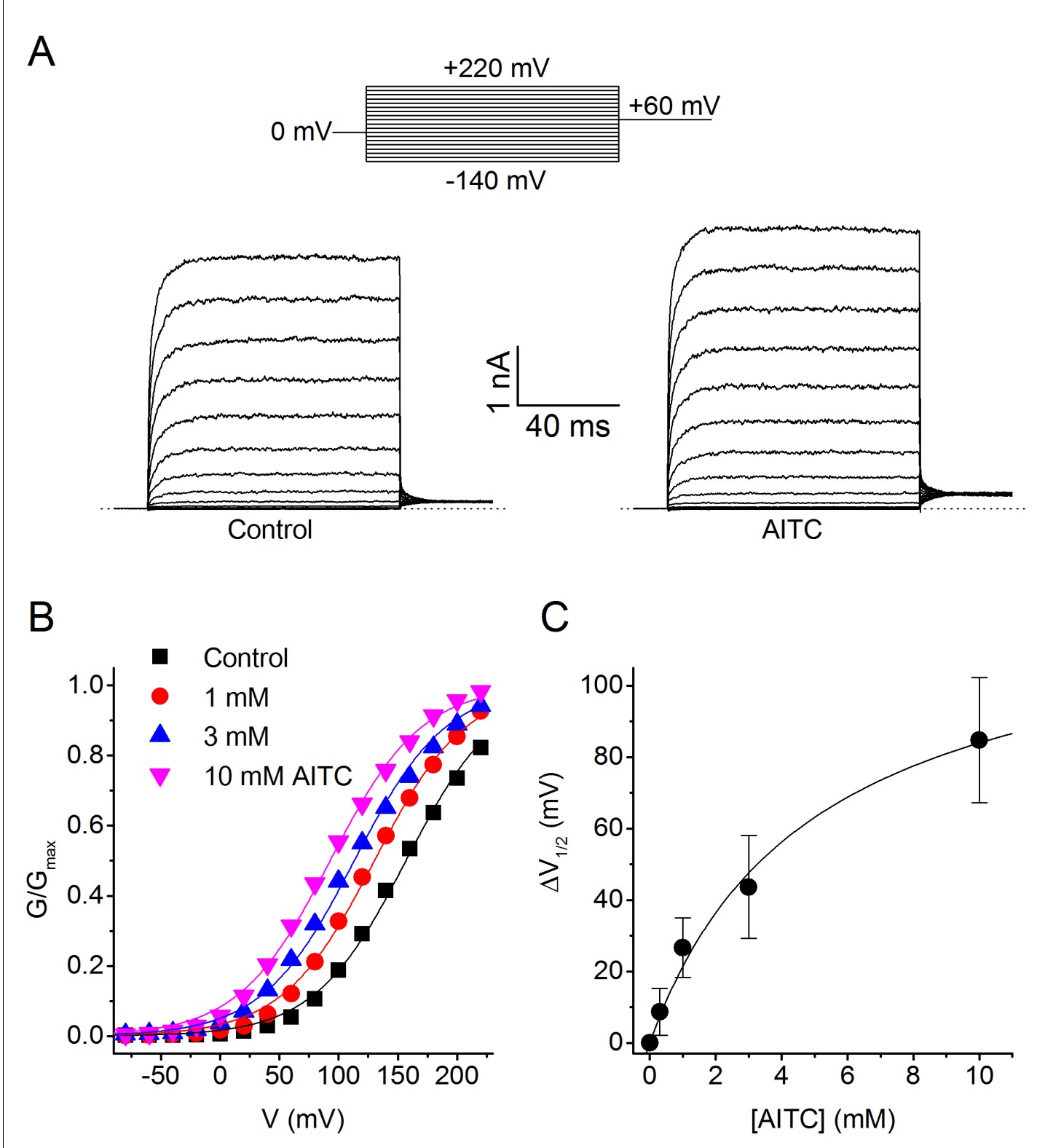

**Figure 3.** Voltage dependence of the activating effect of AITC on human TRPM8. (**A**) TRPM8 currents in response to the indicated voltage step protocol in the absence and presence of AITC (1 mM). (**B**) Voltage-dependent activation curves in control and in the presence of the indicated AITC concentrations, for the cell shown in (**A**). Steady-state conductance (G) was determined as steady-state current divided by test voltage, and normalized to the estimated maximal conductance ($G_{max}$), which was obtained by fitting a Boltzmann function to the curve in the presence of 10 mM AITC. (**C**) Concentration dependence of the shift of voltage-dependent activation curves (n = 7).

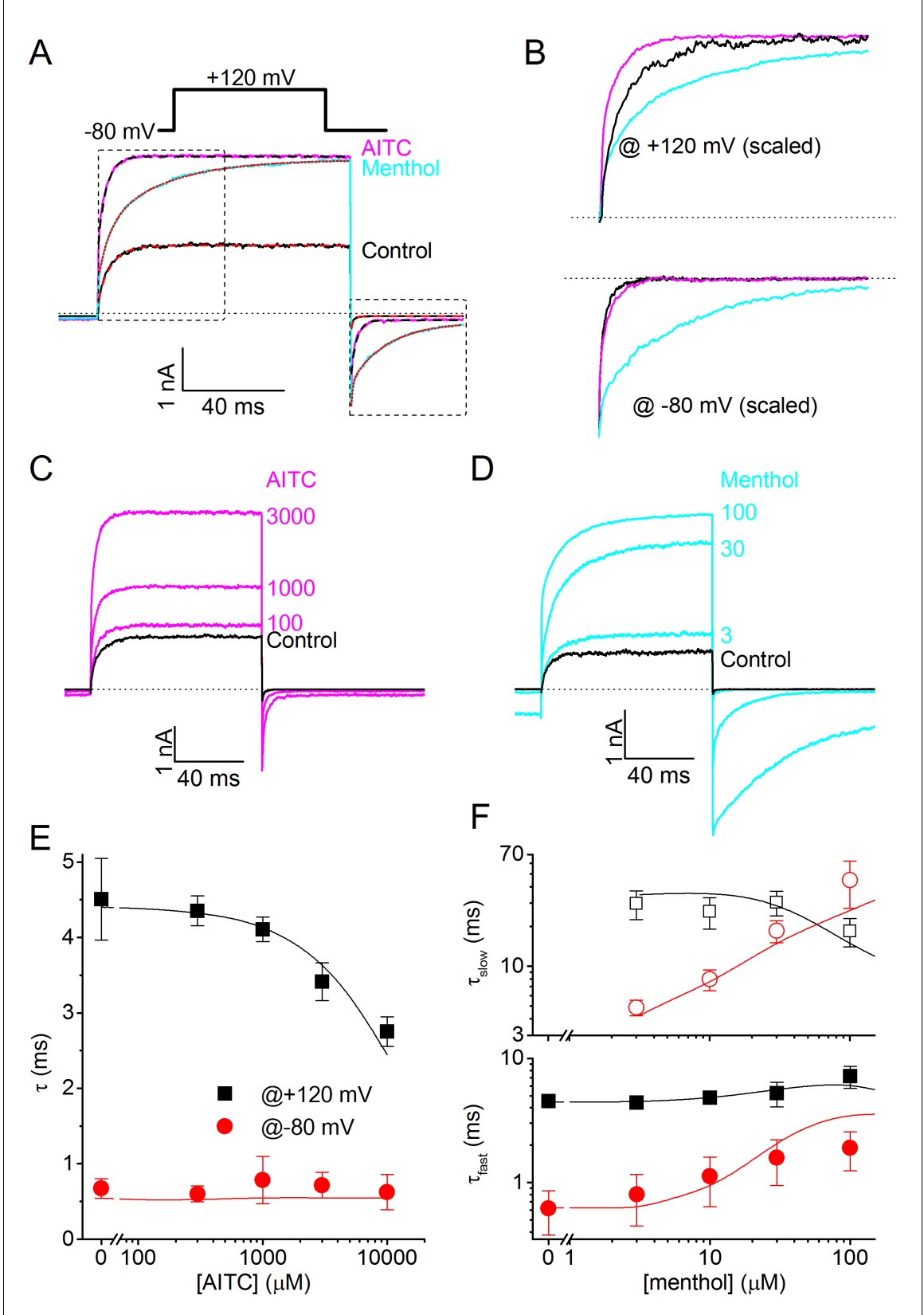

**Figure 4.** Differential effects of AITC and menthol on gating kinetics of human TRPM8. (**A**) Current traces in response to the indicate voltage protocol in control condition and in the presence of menthol (30 μM) and AITC (3 mM). The dashed lines overlaying the control and AITC traces represent single
*Figure 4 continued on next page*

*Figure 4 continued*

exponential fits, the dotted line overlaying the menthol trace represents a double exponential fit. (B) Scaled and expanded currents corresponding to the boxed areas in (A). (C,D) Current traces in response to the voltage protocol from (A) in control condition and the indicated concentrations (in μM) of AITC and menthol. (E) Mono-exponential time constants for current relaxation at +120 and −80 mV in the presence of indicated concentrations of AITC (n = 8). Solid lines represent model predictions, obtained by fitting a mono-exponential function to simulated currents like those shown in *Figure 5D*. (F) Fast and slow exponential time constants for current relaxation at +120 mV and −80 mV in the presence of indicated concentrations of menthol (n = 5). Solid lines represent model predictions, obtained by fitting a double exponential function to simulated currents like those shown in *Figure 5E*. See *Figure 4—figure supplement 1* for more details on the curve fitting.

The following figure supplements are available for figure 4:

**Figure supplement 1.** Mono- and bi-exponential fits of experimental and modeled current relaxation time courses of human TRPM8.

**Figure supplement 2.** Effects of thymol, icilin and linalool on gating kinetics of human TRPM8.

in BK channels were $Ca^{2+}$ binding rates were assumed to be diffusion-limited (*Cox et al., 1997*). Off rates ($k_{off}$) were constrained by the $K_d$ and on rates. Importantly, we obtained excellent fits to the experimental data when we set fixed that menthol binding acts exclusively by stabilization the open state, while AITC acts by destabilization of the closed state (*Figure 5B,C*). Model parameters obtained from the fits are listed in *Table 1*. Gratifyingly, the model accurately predicts the concentration-dependent effects of AITC and menthol on TRPM8, including the mono-exponential time constants in the presence of different AITC concentrations, as well as the bi-exponential relaxation kinetics in the presence of menthol, respectively (*Figure 5D,E*; *Figure 4E,F*; *Figure 4—figure supplement 1*). Based on these results, we propose that AITC represents the first example of a type II TRPM8 agonist, acting primarily by destabilizing the closed channel, which contrasts to the Type I agonists, such as menthol, icilin, thymol and linalool, which primarily stabilize the open channel.

We also tested the combined effect of AITC and menthol on the kinetics of TRPM8 activation and deactivation. In line with the above, application of 50 μM menthol results in slower activation and deactivation kinetics, due to the stabilization of the open state (*Figure 5—figure supplement 2*). Addition of 3 mM AITC in the continued presence of menthol resulted in faster activation kinetics, without affecting the time course of deactivation (*Figure 5—figure supplement 2*). These results are in line with the predictions of the MWC model, assuming that Type I and Type II agonists can act simultaneously and independently, resulting in both stabilization of the open and destabilization of the closed state (*Figure 5—figure supplement 2*).

## Type I versus Type II agonists: effect during action potentials

In the context of a sensory neuron, activation of ion channels such as TRPM8 causes influx of $Na^+$ and $Ca^{2+}$, which depolarizes the membrane and, when the threshold is reached, causes action potential generation (*Vriens et al., 2014*). The differential effects of Type I and Type II agonists on the gating kinetics of TRPM8 suggest that they may have distinct effects on TRPM8-mediated currents and calcium signals during rapid neuronal action potentials. To investigate this possibility, we measured TRPM8 currents evoked by voltage waveforms mimicking action potentials in sensory neurons in the presence of AITC (3 mM) or menthol (30 μM). Note that, at a physiological holding potential of −60 mV, these concentrations resulted in comparable steady-state inward current amplitudes (*Figure 6A*). In response to the action potential waveforms, the current in the presence of AITC mainly manifested during the upstroke phase, and rapidly deactivates upon action potential repolarization. In comparison, in the presence of menthol, the peak outward current is smaller but a more prominent inward TRPM8 current is observed during the repolarization phase of the action potential (*Figure 6A–C*). These differential effects of AITC and menthol on TRPM8 currents during an action potential are fully in line with the predictions of the MWC model for type I versus Type II agonists (*Figure 6A*). We also compared the cumulative influx and efflux of charge during a 1-s train of action potential waveforms at 8 Hz, a typical firing rate of cold-sensitive neurons (*Orio et al., 2012*). As illustrated in *Figure 6D,E*, net charge influx is larger in the presence of menthol, whereas net charge efflux is larger in the presence of AITC.

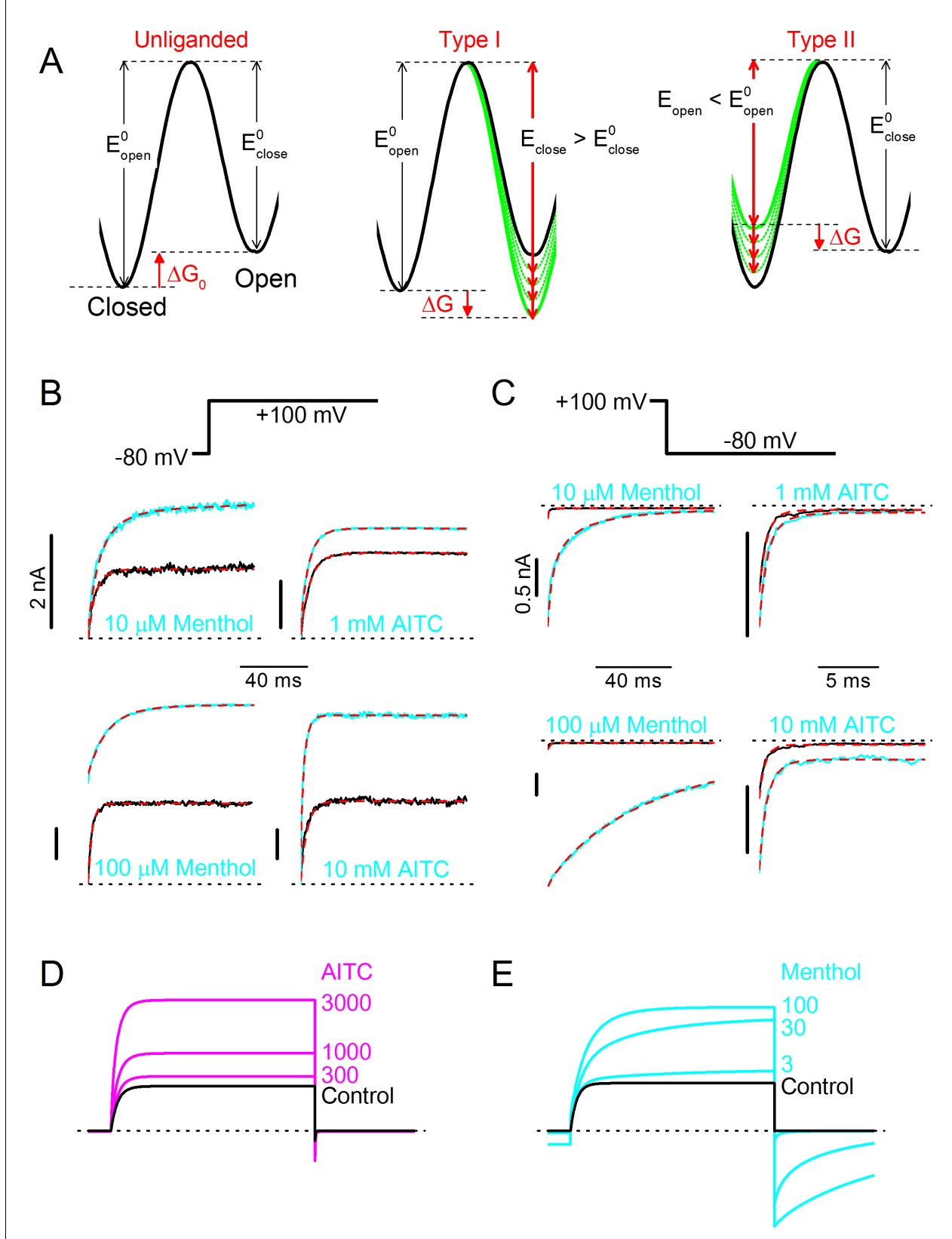

**Figure 5.** Type I (menthol-like) versus Type II (AITC-like) TRPM8 agonists. (**A**) (*left*) Energy diagram for the transition between the closed and open channel conformation in a non-liganded channel. Steady-state equilibrium is determined by $\Delta G_0$, whereas $E_{open}$ and $E_{close}$ determine the opening and
*Figure 5 continued on next page*

*Figure 5 continued*

closing rates, respectively. (*right*) Alteration in the energy profile upon binding of Type I and Type II ligands. The black line represents the non-liganded channel, whereas the green lines represent channels with 1–4 bound ligands. The corresponding kinetic schemes are provided in Supplementary *Figure 2*. (B,C) Activation (B) and deactivation (C) time courses in the absence and presence of the indicated concentrations of menthol or AITC. Overlaid dashed lines represent global fits to the control and ligand-activated current traces. (D,E) Model predictions corresponding to the experimental data shown in *Figure 4C,D*.

The following figure supplements are available for figure 5:

**Figure supplement 1.** Kinetic schemes of the MWC model, depicting the differential effects of Type I and Type II ligands.

**Figure supplement 2.** Combining Type I and Type II agonists.

Under normal physiological conditions, inward TRPM8 current is partially carried by $Ca^{2+}$ ions (*McKemy et al., 2002*; *Peier et al., 2002*). Since our results indicated substantial differences in charge influx during action potentials between Type I and Type II agonists, we expected that menthol and AITC may show differential efficacy in evoking $Ca^{2+}$ transients in excitable *versus* non-excitable cells. To test this, we compared the relative responses to menthol and AITC in TRPM8-expressing mouse sensory neurons *versus* (non-excitable) HEK293 cells heterologously expressing TRPM8. For these latter experiments, we used HEK293 cells transiently expressing the mouse TRPM8 orthologue, and first tested current responses to AITC. Like its human orthologue, mouse TRPM8 was rapidly activated by AITC, and the difference in gating kinetics in the presence of AITC *versus* menthol was also observed (*Figure 7—figure supplement 1*). However, interestingly, AITC-induced current inhibition was much less pronounced in mouse TRPM8 compared to the human orthologue (*Figure 7—figure supplement 2*): at the end of a 60-s application of 3 mM AITC, mouse TRPM8 current amounted to 88 ± 6% (n = 8) of the peak current, compared to 21 ± 5% in the case of human TRPM8 (n = 7; p=0.00004). A further analysis of this species difference in AITC-induced inhibition is provided in *Figure 7—figure supplement 2*.

To specifically analyze TRPM8-mediated responses to AITC and menthol in mouse sensory neurons, we used TRPA1/TRPV1 double knockout mice, only examined cells that showed robust responses to both agonists, and controlled that these responses were fully inhibited in the presence of AMTB, as outlined in *Figure 1*. In these cells, we found that the amplitudes of $Ca^{2+}$ transients evoked by a 60-s-long applications of 3 mM AITC were on average ~30% smaller than those evoked by 30 μM menthol (*Figure 7A,D*). Likewise, the peak rate of calcium rise, which represents a

**Table 1.** Experimentally derived model parameters describing the action of menthol and AITC on TRPM8 gating.

| Parameter | Value | Source |
|---|---|---|
| z | 0.82 | (*Voets et al., 2007*) |
| $\Delta\Delta G_{AITC}$ | −2.7 ± 0.4 kJ mol$^{-1}$ | Steady-state activation curves (n = 7) |
| $\Delta\Delta G_{menthol}$ | −4.5 ± 0.4 kJ mol$^{-1}$ | Steady-state activation curves (n = 6) |
| $K_{d,AITC}$ | 2.9 ± 0.6 mM | Steady-state activation curves (n = 7) |
| $K_{d,menthol}$ | 21 ± 4 μM | Steady-state activation curves (n = 6) |
| $\alpha_0(0)$ | 10.4 ± 1.2 s$^{-1}$ | Global kinetic fit (n = 14) |
| $\beta_0(0)$ | 1.11 ± 0.15 ×10$^3$ s$^{-1}$ | Global kinetic fit (n = 14) |
| $k_{on,AITC}$ | 95 ± 35 ×10$^3$ M$^{-1}$ s$^{-1}$ | Global kinetic fit (n = 7) |
| $k_{on,menthol}$ | 551 ± 210 ×10$^3$ M$^{-1}$ s$^{-1}$ | Global kinetic fit (n = 7) |

Displayed are values for the different parameters that determine the MWC model. For the global kinetic fits, cells were included for which current traces were fit at minimally tree ligand concentrations and two voltages. More details are provided in the text.

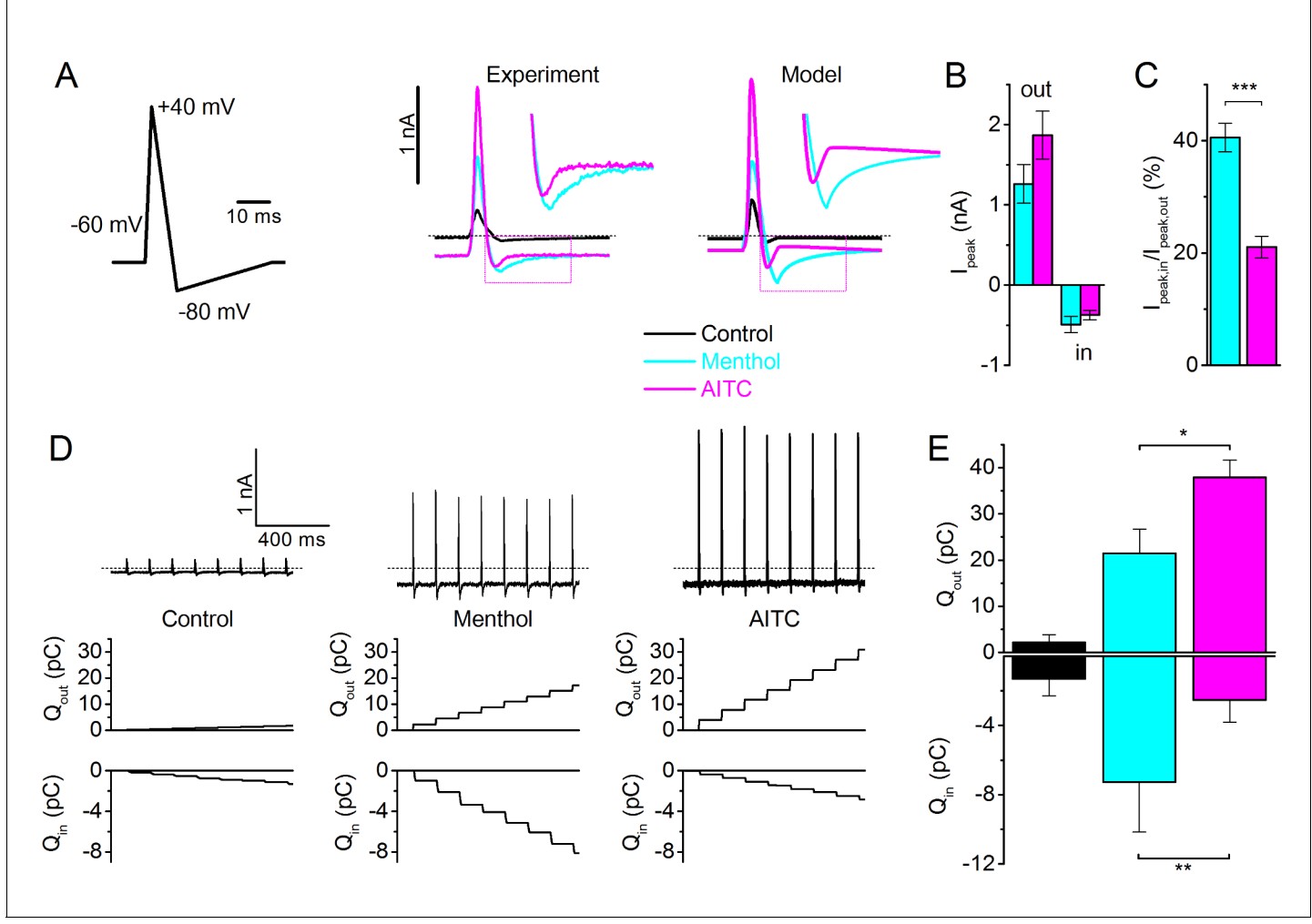

**Figure 6.** TRPM8 gating during an action potential – Type I versus Type II ligands. (**A**) Voltage protocol simulating a sensory neuron action potential (*left*); TRPM8 currents in HEK293 cells in response to the action potential waveform in control condition and during application of menthol (30 µM) and AITC (3 mM) (*middle*); and corresponding model simulation (*right*). Boxed areas are expanded in the inset. (**B**) Peak outward and inward currents during the action potential waveform in the presence of menthol (cyan) and AITC (magenta; n = 6). (**C**) Ratio between peak inward and peak outward current in the presence of menthol or AITC. ***p<0.001. (**D**) TRPM8 current responses during a train of action potentials (1 s; 8Hz) in control condition and during application of menthol and AITC (*top*); outward (*middle*) and inward (*bottom*) charge displacement during the action potential train, determined as the integrated current after subtraction of the holding current. (**E**) Mean inward and outward charge displacement for the two ligands (n = 5). *p<0.05; **p<0.01.

measure for the maximal inward calcium current, was consistently smaller in response to AITC than to menthol in neurons (***Figure 7A,D***). Interestingly, we observed an opposite potency of the same concentrations of menthol and AITC in HEK cells expressing mouse TRPM8: AITC evoked larger calcium increases and with a higher peak rate of calcium rise than did menthol (***Figure 7B,D***). Moreover, if action potential firing in the TRPA1/TRPV1-deficient sensory neurons was blocked using tetrodotoxin (TTX; 1 µM), we found a similar ratio of AITC *versus* menthol responses as in HEK cells (***Figure 7C,D***), with AITC being slightly more potent than menthol. Taken together, these data provide further support for the notion that, compared to Type II agonists (e.g. AITC), Type I agonists (e.g. menthol) are more potent in evoking calcium influx in excitable cells, due to enhanced calcium influx during the prolonged inward tail currents following action potentials. In cells that do not fire action potentials, rapid changes in membrane potential are not expected, and hence the kinetic differences between the two types of agonists will not affect calcium signals.

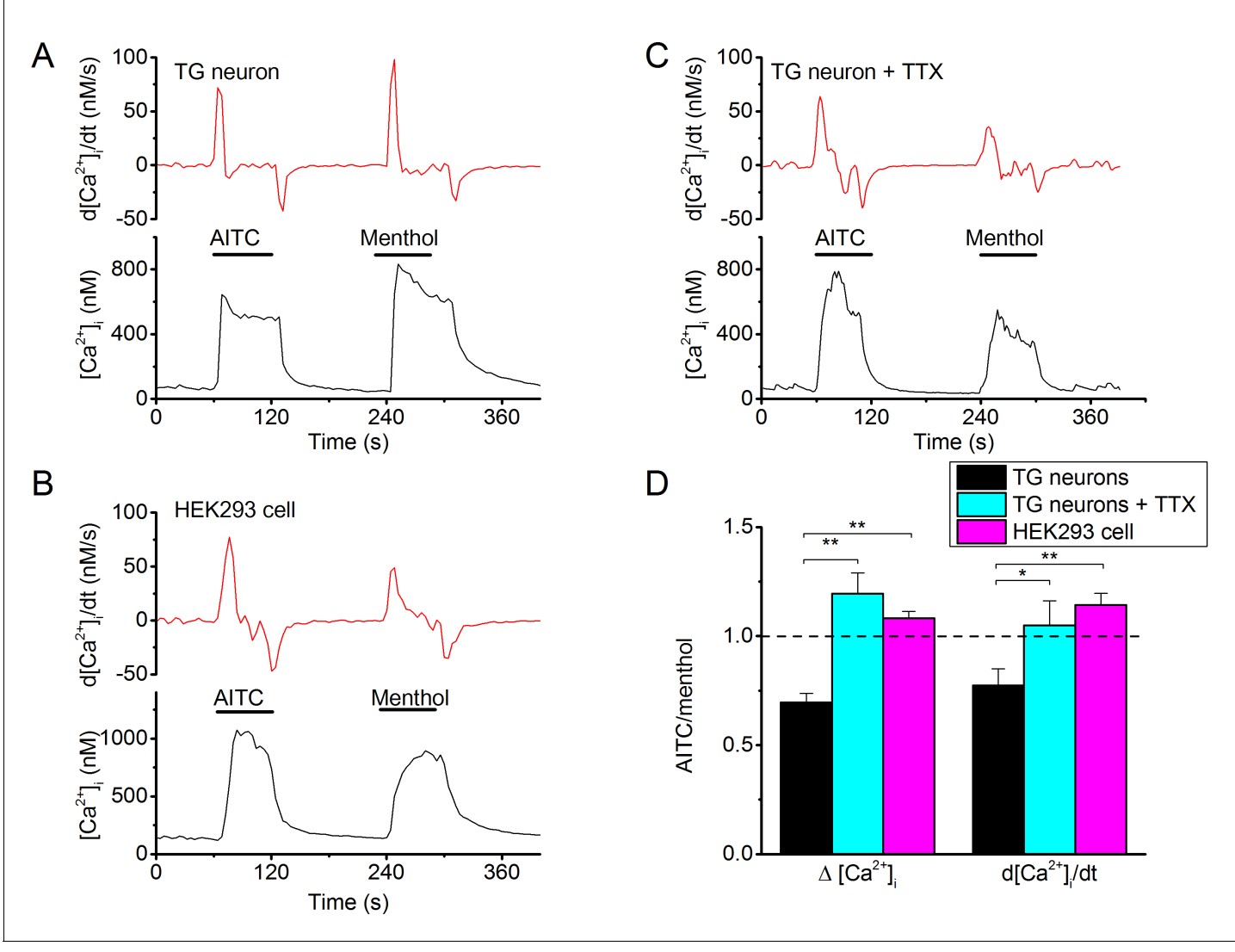

**Figure 7.** Differential effectiveness of Type I and Type II agonists in excitable versus non-excitable cells. (**A**) Fura-2-based intracellular calcium measurements in mouse trigeminal neurons from TRPV1/TRPA1 double knockout mice showing increases in intracellular calcium in response to AITC (3 mM) and menthol (30 μM). The upper trace shows the time differential of the intracellular calcium concentration, which represents a measure of net calcium influx/extrusion mechanisms. The TRPM8-dependence of the responses was ensured based on full block by AMTB (as in **Figure 1**; not shown). (**B**) Same as (**A**), but in the presence of TTX (1 μM) to block neuronal action potentials. (**C**) Same as (**A**), but in a HEK293 cell expressing mouse TRPM8. Non-transfected cells did not show any detectable response to AITC or menthol. (**C**) Relative stimulatory effect of menthol and AITC in control trigeminal neurons (n = 81 from 9 different mice), trigeminal neurons treated with 1 μM TTX (n = 3 from 3 different mice) and HEK293 cells (n = 448). *, **, ***: $p < 0.05$, 0.01 and 0.001, respectively, in paired t-test comparing the response to AITC and menthol within individual cells. ###: $P < 0.001$ in unpaired t-tests comparing TG neurons and HEK293 cells.

The following figure supplements are available for figure 7:

**Figure supplement 1.** Activation of mouse TRPM8 by AITC.

**Figure supplement 2.** AITC-induced current inhibition in human *versus* mouse TRPM8, as well as in chimeric channels.

# Discussion

While there are already numerous natural and synthetic agonists known for TRPM8 (*Almaraz et al., 2014*), our results demonstrate that AITC is an atypical agonist, with a mode of action that is fundamentally different from that of all other known TRPM8-activating stimuli. Activation of TRPM8 by

cooling or by known agonists such as the natural compounds menthol, thymol and linalool, and the synthetic agonists such as icilin and halothane, is associated with a slowing of the kinetics of voltage-dependent channel gating (*Vanden Abeele et al., 2013*; *Voets et al., 2004*; *Voets et al., 2007*). This slowing of the gating kinetics can be directly explained by a stabilization of the open channel relative to the transition state, as we illustrated in this work for menthol and elsewhere for cooling (*Voets et al., 2004*). In clear contrast, activation of TRPM8 by AITC resulted in an acceleration of the kinetics of voltage-dependent gating, and we show here that this can be fully explained by a mechanism where AITC leads to a relative destabilization of the closed conformation relative to the transition state. Based thereon, we propose that TRPM8 agonists can be classified as either Type I, causing a relative stabilization the open state, or Type II, causing a relative destabilization the closed state, and we provide a kinetic voltage-dependent Monod-Wyman-Changeux-type model that faithfully reproduces their differential agonist effects. Such classification may also be extended to activating ligands of other voltage- and ligand-sensitive TRP channels. For instance, published current traces suggest that activation of TRPV1 by capsaicin or low pH is associated with faster activation time courses upon depolarization (*Aneiros et al., 2011*; *Voets et al., 2004*), classifying them as Type II ligands, whereas the activating effects of phosphatidylinositol-4,5-bisphosphate ($PIP_2$) or lyso-phosphatidic acid are associated with slower activation and longer deactivating tails upon repolarization, classifying them a Type I agonists (*Nieto-Posadas et al., 2012*; *Ufret-Vincenty et al., 2015*).

The classification of activating ligands as either Type I or Type II is useful for several purposes. First, this information can provide important insights into ligand-induced structural rearrangements during channel gating, and may help interpreting ligand-bound channel structures. Indeed, our results indicate that Type II ligands such as AITC reduce the difference Gibbs free energy between the closed state and the transition state ($\Delta G^{\alpha\dagger}$), without affecting the difference Gibbs free energy between the open state and the transition state ($\Delta G^{\beta\dagger}$). This suggests that the AITC-induced conformational change at its binding site occurs early in the gating process, prior to the main close-open transition. Oppositely, Type I ligands such as menthol cause an increase in $\Delta G^{\beta\dagger}$, without affecting $\Delta G^{\alpha\dagger}$. This indicates that the menthol-induced conformational change at its binding sites occurs later than the main close-open transition. This analysis and interpretation is reminiscent of the rate-equilibrium free-energy relationship (REFER) approach, which has used to evaluate the effects of perturbations (e.g. ligands or mutations) on the equilibrium of reactions, including the gating of ion channels such as the nicotinic acetylcholine receptor and CFTR (*Grosman et al., 2000*; *Sorum et al., 2015*). In REFER, the effect of a family of perturbations on channel gating is quantified using the φ-value, which is the slope of a plot of the logarithm of the opening rate (log α) versus the log of the gating equilibrium constant (log $K_{eq}$), where $K_{eq}$ is the ratio of the opening (α) and closing (β) rate ($K_{eq} = \alpha/\beta$) (*Auerbach, 2007*). In the case of type I ligands, ligand binding affects the equilibrium solely by decreasing β, yielding φ = 0. Following the REFER theorem, this indicates late movement (*Auerbach, 2007*). Type II ligands affect the equilibrium entirely by affecting α, yielding φ = 1, indicating early movement. We also found that the effects of simultaneously applied menthol and AITC on channel gating are well described assuming independent binding and additive effects on the gating equilibria, which further supports the notion that type I and type II ligands act at distant binding sites with different timing for conformational changes.

Second, based on our kinetic fits, we obtained estimates for the on rates for ligand binding to TRPM8. For example, for menthol binding we obtained a $k_{on}$ of 0.55 $\mu M^{-1}s^{-1}$, which is well below the diffusion-limited rate (>100 $\mu M^{-1}s^{-1}$), and also one order of magnitude or more lower than binding rates for ligands to synaptic ligand-gated channels, such as the ionotropic receptors for glutamate ($k_{on} \approx$ 5 $\mu M^{-1}s^{-1}$) (*Clements and Westbrook, 1991*), ATP ($k_{on} \approx$ 12 $\mu M^{-1}s^{-1}$) (*Bean, 1990*) or acetylcholine ($k_{on} \approx$ 60 $\mu M^{-1}s^{-1}$) (*Sine et al., 1990*). The relatively slow ligand equilibration kinetics for menthol are in line with the distinct structural properties of ligand binding sites in TRP channels compared to these classical ionotropic receptors. Indeed, whereas binding sites for glutamate, ATP and acetylcholine are located extracellularly (*Hille, 2001*), directly accessible from the aqueous phase, the binding site for menthol is located in a hydrophobic domain in between the transmembrane helices (*Bandell et al., 2006*; *Voets et al., 2007*). The observation that current relaxation time courses of TRPM8 in the presence of menthol become multi-exponential is then a direct consequence of the slow equilibration rate of menthol with its binding site in comparison to the transition rates between closed and open channel conformations.

Finally, we showed that differential effect on voltage-dependent gating of Type I and II agonists is reflected in distinctive TRP channel-mediated currents during rapid changes in membrane voltage, for instance during an action potential in a sensory neuron. Indeed, as predicted by our model, the AITC-induced TRPM8 current during a typical action potential waveform mainly manifest during the upstroke phase, and rapidly deactivates upon action potential repolarization. In contrast, an equipotent concentration of menthol (i.e. a concentration of menthol provoking a similar steady-state current) results in a less outward current but more prominent activation of inward TRPM8 current during the repolarization phase following an action potential, and thus leads to more $Ca^{2+}$ influx via TRPM8. In line herewith, we found that inhibiting action potential firing using TTX has a more profound effect on menthol-induced responses than on AITC-induced responses in sensory neurons. These findings illustrate the importance of evaluating the mode of action of ligands on voltage-dependent TRP channels, especially when extrapolating results from non-excitable heterologous expression systems to physiological effects in excitable cells such as neurons, cardiomyocytes or pancreatic beta cells.

Using the voltage-dependent Monod-Wyman-Changeux-type model, we assumed that for any number of bound ligands the transition between closed and open channel conformation is a one-step process, determined by forward and backward rates $\alpha_i$ and $\beta_i$. Whereas this assumption is in line with the mono-exponential kinetics we generally observed in our experiments in the absence of ligands (*Voets et al., 2004*), it is probably a simplification of the full gating intricacies of TRPM8, and models with one or more closed-closed transitions preceding channel opening have been proposed (*Fernandez et al., 2011*; *Raddatz et al., 2014*). Nevertheless, even when using such more complex models, the ligands' effects on TRPM8 gating kinetics can only be explained assuming that Type II ligands cause acceleration of the gating transition(s) towards the open state, whereas Type I ligands slow down the backward rate(s) from the open state.

Our results demonstrate that TRPM8 underlies the residual TRPA1- and TRPV1-independent responses to AITC in mouse sensory neurons. Under our experimental conditions, activation of TRPM8 by AITC only occurred in the high micromolar to millimolar concentration range. As such, TRPM8 is about two orders of magnitude less sensitive to AITC than TRPA1, for which concentrations for half-maximal activation of 5–50 μM have been reported (*Bandell et al., 2004*; *Everaerts et al., 2011*; *Jordt et al., 2004*), but comparable to TRPV1, for which a concentration for half-maximal activation of 3 mM was found at room temperature (*Everaerts et al., 2011*). These findings are in line with in vitro experiments in sensory neurons, showing that AITC concentrations ≤100 μM evoke responses that are strictly TRPA1-dependent (*Bautista et al., 2006*; *Everaerts et al., 2011*), whereas higher concentrations can also evoke TRPA1-independent responses mediated by TRPV1 or TRPM8. AITC is extensively used in in vivo experiments to induce pain and inflammation (*Julius, 2013*). In such assays, experimental solutions that are injected or topically applied typically contain AITC at concentrations between 10 and 100 mM (*Bautista et al., 2006*; *Caterina et al., 2000*). Whereas earlier studies have clearly shown that pain and inflammatory responses under such experimental conditions are largely mediated by TRPA1 and TRPV1 (*Bautista et al., 2006*; *Everaerts et al., 2011*; *Kwan et al., 2006*), our present results suggest that also TRPM8-positive sensory nerve endings may become activated at these AITC doses. Since activation of TRPM8-expressing neurons can cause analgesia in animal models of acute and chronic pain (*Liu et al., 2013*; *Proudfoot et al., 2006*), the effects of AITC on TRPM8 that we describe here need to be taken into account when using AITC as a proalgesic and/or proinflammatory agent. TRPM8 may contribute to the complex psychophysical effects that one experiences upon eating spices containing millimolar concentrations of AITC such as mustard or wasabi (*Nilius and Appendino, 2013*). In line herewith, a transient increase in cold sensitivity was observed upon application of 100 mM AITC on the tongue of human volunteers (*Albin et al., 2008*). Although speculative, this may correlate with the transient activation followed by channel inhibition that we observed in human TRPM8.

In voltage-gated ion $Na^+$, $K^+$ and $Ca^{2+}$ channels, ligand modulators have since decades been classified based on their distinct state-dependent effects on channel gating (*Hille, 2001*), and this mechanistic insight has been key to understanding their physiological impact in for instance neurons and cardiac cells (*Sack and Sum, 2015*). In this study, we demonstrate for the first time the existence of two types of agonists with distinct state-dependent effects for a member of the TRP superfamily, the cold-sensitive TRPM8, and provide a paradigm for their differential effects in sensory neurons. We argue that establishing the state-dependent mode of action of (ant)agonists of this and other TRP

channels will be essential to clarify their physiological actions as well as to understand their impact on conformational changes in the channel molecule.

## Materials and methods

### Cells and transfection

HEK293 were grown in DMEM containing 10% (v/v) fetal calf serum, 4 mM L-alanyl-L-glutamine, 100 U ml$^{-1}$ penicillin and 100 µg ml$^{-1}$ streptomycin at 37°C in a humidity controlled incubator with 10% $CO_2$. For patch-clamp and calcium imaging, cells were transiently transfected with different human (NM024080) or mouse (NM134252) TRPM8 constructs cloned in the bicistronic pCAGGSM2-IRES-GFP vector using TransIT-293 transfection reagent (Mirus, Madison, WI). Mutations and chimeras were made using the PCR-overlap technique, and verified by Sanger sequencing (LGC-genomics, Germany). Chimeras were made by swapping the N termini (amino acids 1–336) or C termini (amino acids 993–1004) between the orthologues. For TIRF imaging, we used human TRPM8 linked to mCherry at its C-terminal end (*Ghosh et al., 2016*).

Trigeminal ganglia (TGs) of 10-16-week-old female *Trpv1$^{-/-}$/Trpa1$^{-/-}$* mice were isolated after $CO_2$ euthanasia. Bilateral TGs were collected and digested with 1 mg/ml collagenase and 2.5 mg/ml dispase dissolved in 'basal medium' (Neurobasal A medium supplemented with 10% FCS) (all from Gibco/Life Technologies, Belgium) at 37°C for ca. 45–60 min. Digested ganglia were gently washed once in 'basal medium' and twice in 'complete medium' (Neurobasal A medium supplemented with 2% B27 [Invitrogene/Life Technologies, Belgium], 2 ng/ml GDNF [Invitrogen/Life Technologies] and 10 ng/ml NT4 [Peprotech, UK]) and mechanically dissociated by mixing with syringes fitted with increasing needle gauges. Neurons were seeded on poly-L-ornithine/laminin-coated glass bottom chambers (Fluorodish WPI, UK) and cultured at 37°C in complete medium overnight. These experiments were approved by the KU Leuven Ethical Committee Laboratory Animals under project number P192/2014.

### Patch-clamp

Between 16 and 24 hr after transfection, currents were recorded in the whole-cell or inside-out configurations of the patch-clamp technique using an EPC-9 amplifier and PULSE software (HEKA Elektronik, Germany). Data were sampled at 5–20 kHz and digitally filtered off-line at 1–5 kHz. In the whole-cell mode, between 70 and 90% of the series resistance was compensated, and recordings where the estimated voltage error due to uncompensated series resistance exceeded 10 mV were excluded from analysis. Whole-cell recordings were performed using an intracellular solution containing (in mM) 150 NaCl, 5 MgCl$_2$, 5 EGTA and 10 HEPES, pH 7.4. The extracellular solution contained (in mM) 150 NaCl, 1 MgCl$_2$ and 10 HEPES, pH 7.4. In inside-out recordings, the extracellular solution was used as pipette solution, and ligands were included in the intracellular bath solution.

### Calcium imaging

For intracellular $Ca^{2+}$ measurements, cells were incubated with 2 µM Fura-2 acetoxymethyl ester for 30 min at 37°C. The fluorescent signal was measured during alternating illumination at 340 and 380 nm using either an Cell$^M$(Olympus, Belgium) or Eclipse Ti (Nikon, Belgium) fluorescence microscopy system. The standard extracellular solution used in ratiometric $[Ca^{2+}]_i$ measurements contained (in mM) 150 NaCl, 5 KCl, 2 CaCl$_2$, 1.5 MgCl$_2$, and 10 HEPES, pH 7.4.

### TIRF imaging

TIRF images were acquired using a through-the-lens TIRF system that was built around an inverted Axio Observer.Z1 microscope equipped with a X-100 oil objective numerical aperture (NA)=1.45 (Zeiss, Germany), a Orca-R$^2$ camera (Hamamatsu, Japan), and using a 561-nm laser. Time series of images at 1-s intervals were recorded. Constant focus was maintained using the Definite Focus module (Zeiss). The TIRF angle was set to achieve an evanescent field with a characteristic penetration depth (i.e., the distance in the z direction over which the intensity declines e-fold) of 90 nm. Cells on 25-mm glass coverslips were placed in a custom-made chamber and imaged at 25°C.

## Chemicals

Chemicals were obtained from Sigma (Belgium), unless indicated otherwise. AITC, menthol, thymol, linalool were dissolved in ethanol to obtain 1-M stock solutions. Icilin was dissolved in DMSO to obtain a 50-mM stock solution. Tetrodotoxin (TTX; from Alomone labs, Israel) was dissolved in acetate buffer at a concentration of 31 mM.

## Modeling and fitting

As a starting point to model the gating of TRPM8 in the absence and presence of ligands, we built on our earlier work describing the effects of temperature and menthol on steady-state TRPM8 currents (*Janssens and Voets, 2011*; *Voets, 2012*; *Voets et al., 2004*, *2007*). In the absence of ligands, the transition between the closed and open conformation of the channel is determined by the opening and closing rates:

$$\alpha_0 = \kappa \frac{k_b T}{h} e^{-\frac{\Delta G^{\alpha\dagger}}{RT}} \tag{1}$$

and

$$\beta_0 = \kappa \frac{k_b T}{h} e^{-\frac{\Delta G^{\beta\dagger}}{RT}}, \tag{2}$$

where $k_b$ is the Boltzmann constant ($1.38 \times 10^{-23}$ J K$^{-1}$), $T$ the absolute temperature, $h$ the Planck constant ($6.63 \times 10^{-34}$ J s), R the universal gas constant ($8.314 \times$ J K$^{-1}$ mol$^{-1}$) and $\kappa$ the transmission coefficient, whose value for the studied processes is unknown. $\Delta G_0^{\alpha\dagger}$ ($\Delta G_0^{\beta\dagger}$) represents the difference in free energy between closed (open) state and the transition state of the non-liganded channel (see *Figure 5*), and depend on temperature and voltage (*V*) according to:

$$\Delta G^{\alpha\dagger} = \Delta H^{\alpha\dagger} - T\Delta S^{\alpha\dagger} - 0.5zFV \tag{3}$$

and

$$\Delta G^{\beta\dagger} = \Delta H^{\beta\dagger} - T\Delta S^{\beta\dagger} + 0.5zFV. \tag{4}$$

$\Delta H_0^{\alpha\dagger}$ and $\Delta H_0^{\beta\dagger}$ represent the differences in enthalpy and $\Delta S_0^{\alpha\dagger}$ and $\Delta S_0^{\beta\dagger}$ the differences in entropy between, respectively, the closed and open state and the transition state, $z$ the gating charge, and $F$ the Faraday constant (96485 C mol$^{-1}$). In our experiments, temperature was kept constant at 23°C, yielding:

$$\Delta G^{\alpha\dagger} = \Delta G_{0\,mV}^{\alpha\dagger} - 0.5zFV \tag{5}$$

and

$$\Delta G^{\beta\dagger} = \Delta G_{0\,mV}^{\beta\dagger} + 0.5zFV. \tag{6}$$

In the absence of ligands, the voltage-dependent opening and closing rates are then given by:

$$\alpha_0(V) = \alpha_0(0) \times e^{\frac{0.5zFV}{RT}} \tag{7}$$

and

$$\beta_0(V) = \beta_0(0) \times e^{\frac{-0.5zFV}{RT}}, \tag{8}$$

where $\alpha_0(0)$ and $\beta_0(0)$ represent the opening and closing rates at 0 mV.

As evidenced in earlier work (*Janssens and Voets, 2011*), we consider that TRPM8 has 4 independent and energetically equivalent ligand binding sites (i.e. one per subunit), with an affinity $K_d$ of the open channel determined by ligand-channel association and dissociation rates $k_{on}$ and $k_{off}$ ($K_d = k_{off}/k_{on}$). The energetic effect of ligand binding on steady-state channel equilibrium can be quantified as $\Delta\Delta G_{ligand}$, which represents the change of the difference in Gibbs free energy between the closed and open state of the channel ($\Delta G$) upon binding of one ligand molecule to one of the four subunits. Values for $K_{d,menthol}$, $\Delta\Delta G_{menthol}$, $K_{d,AITC}$ and $\Delta\Delta G_{AITC}$ were obtained from concentration-

dependent changes in the midpoint of the steady-state voltage-dependent activation curves ($\Delta V_{1/2}$), according to:

$$\Delta V_{1/2} = -\frac{RT}{zF} \ln \frac{\left(1 + \frac{[L]}{K_d}\right)^4}{\left(1 + \frac{[L]}{K_d} \times \exp\frac{\Delta\Delta G}{RT}\right)^4}.$$ (9)

Since saturating effects of AITC could not be obtained at the highest concentration tested (10 mM; on the limits of solubility), values for $K_{d,AITC}$ and $\Delta\Delta G_{AITC}$ should be considered as approximate.

In the presence of a Type I agonist such as menthol, ligand binding stabilizes the open state, without affecting the closed or transition states (*Figure 5* and *Figure 5—figure supplement 1*). Therefore, the opening rate of a channel with *i* bound ligands remains unaltered

$$\alpha_i(V) = \alpha_0(V),$$ (10)

where as the closing rate becomes slower for each bound ligand

$$\beta_i(V) = \beta_0(V) \times e^{-\frac{i \times \Delta\Delta G_{ligand}}{RT}}.$$ (11)

In the presence of a Type II agonist such as AITC, ligand binding destabilizes the closed state, without affecting the open or transition states (*Figure 5—figure supplement 2*). Therefore, the closing rate of the channel remains unaltered

$$\beta_i(V) = \beta_0(V),$$ (12)

where as the opening rate becomes faster for each bound ligand

$$\alpha_i(V) = \alpha_0(V) \times e^{\frac{i \times \Delta\Delta G_{ligand}}{RT}}.$$ (13)

Procedures were written in Igor Pro 6.22 (Wavemetrics, Lake Oswego, OR) to numerically solve the set of 10 differential equations describing the transitions between the 10 states of the model at different voltages and in the presence of different ligand concentrations. Briefly, to fit the gating behavior during voltage steps, eigenvalues and corresponding eigenvectors of the transition matrix were numerically solved using the *MatrixEigenV* operation, and used to calculate the sums of exponential terms describing the time-dependent changes of the probabilities that the channel is in one of the 10 states. The *FuncFit* and *DoNewGlobalFit* procedures were then used to find the model parameters that yield the best global fit to current relaxation time courses measured within one cell in the absence and presence of ligand (*Table 1*). The global kinetic fit included three free parameters: $\alpha_0(0)$, $\beta_0(0)$ and $k_{on}$. Prior to the kinetic fit, $K_{d,ligand}$ and $\Delta\Delta G_{ligand}$ were determined from steady-state currents, according to *Equation 9*; $k_{off}$ was set as $K_d \times k_{on}$; z was fixed at a value of 0.82, based on earlier work (*Voets et al., 2007*). We further assumed that the rate of ligand binding to the open and closed state of the channel were identical, whereas the rate of ligand unbinding from the closed state was constrained by detailed balance. The *integrateODE* operation was used to model TRPM8 currents during voltage steps or action potential waveforms, using mean parameters obtained from the fits (*Table 1*).

## Statistics

Data analysis was performed using Origin 9.0 (OriginLab Corporation, Northampton, MA). Group data are presented as mean ± SEM from *n* cells. Comparison between two groups was done using Student's unpaired or paired test, as indicated. No explicit power analysis was performed prior to the experiments to determine sample size, since we had no means to reliably estimate the size and variability of the effects of the ligands on parameters of TRPM8 gating. For patch-clamp experiments on HEK cells, typically 5–10 cells were measured for each condition, thereby limiting the SEM to ≤20% of the mean value for the relevant parameters. For the calcium imaging experiments on mouse TG neurons, a maximal number of neurons from nine mice isolated on 5 independent days were analyzed. Since highly significant results were obtained from this set of experiments, no further animals were sacrificed.

## Acknowledgements

We acknowledge all members of the Laboratory of Ion Channel Research for helpful discussions. The research leading to these results has received funding from the People Programme (Marie Curie Actions) of the European Union's Seventh Framework Programme (FP7/2007-2013) under REA grant agreement n° 330489, and was further supported by grants from the Belgian Federal Government (IUAP P7/13), the Hercules Foundation (AKUL-029), the Research Foundation-Flanders (G.0565.07), and the Research Council of the KU Leuven (PF-TRPLe).

## Additional information

### Funding

| Funder | Grant reference number | Author |
| --- | --- | --- |
| Onderzoeksraad, KU Leuven | PF-TRPLe | Rudi Vennekens<br>Thomas Voets |
| Federaal Wetenschapsbeleid | IUAP P7/13 | Rudi Vennekens<br>Karel Talavera<br>Thomas Voets |
| Fonds Wetenschappelijk Onderzoek | G.0565.07 | Thomas Voets |
| Seventh Framework Programme | FP7/2007-2013 | Thomas Voets |
| Hercules Foundation | AKUL-029 | Thomas Voets |

The funders had no role in study design, data collection and interpretation, or the decision to submit the work for publication.

### Author contributions

AJ, MG, JV, Conception and design, Acquisition of data, Drafting or revising the article; BIT, Acquisition of data, Drafting or revising the article; DG, Acquisition of data, Analysis and interpretation of data; MM, Acquisition of data, Analysis and interpretation of data, Drafting or revising the article; RV, KT, Conception and design, Drafting or revising the article; TV, Conception and design, Acquisition of data, Analysis and interpretation of data, Drafting or revising the article, Contributed unpublished essential data or reagents

### Author ORCIDs

Thomas Voets, http://orcid.org/0000-0001-5526-5821

### Ethics

Animal experimentation: All experiments were approved by the KU Leuven Ethical Committee Laboratory Animals under project number P192/2014.

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
