## [Decision Letter]

Thank you for submitting your article "Definition of two agonist types at the cold-activated channel TRPM8" for consideration by *eLife*. Your article has been reviewed by three peer reviewers, Kenton J Swartz (Reviewer #1), Jorg Grandl (Reviewer #2) and László Csanády (Reviewer #3), and the evaluation has been overseen by Kenton Swartz as Reviewing Editor and Richard Aldrich as the Senior Editor.

The reviewers have discussed the reviews with one another and the Reviewing Editor has drafted this decision to help you prepare a revised submission.

Summary:

This comprehensive study compares the effects of menthol and allyl isothiocyanate (AITC) on gating of TRPM8 channels. The effects of the two agonists are compared using a broad range of experimental approaches, including kinetic analysis of current responses to voltage steps, analysis of charge flow during action potentials evoked by voltage clamp, and observation of time courses of intracellular [Ca^2+^] in excitable and non-excitable cells. The key findings are that (i) AITC and menthol responses are strictly correlated in TRPV1/TRPA1-deficient neurons suggesting TRPM8 as a novel AITC target, (ii) AITC induces rapid activation followed by slow inactivation of heterologously overexpressed TRPM8, (iii) AITC-induced activation is rapidly, whereas inactivation is slowly reversible, (iv) menthol and AITC activate TRPM8 through distinct kinetic mechanisms, (v) these differential kinetic mechanisms of the two agonists result in differential rates of Ca^2+^ entry, depending on the cell type (excitable vs. non-excitable).

The most important novel aspect of the study is the detailed analysis of the kinetic mechanisms of menthol and AITC, interpreted in the framework of a Monod-Wyman-Changeux (MWC) type mechanism, which integrates voltage-, and agonist-dependence of the channel. Exposure to AITC is shown to accelerate current relaxations in response to depolarizing, but not repolarizing, voltage steps. In contrast, menthol exposure slows current relaxations in response to voltage steps in both directions. Moreover, whereas current relaxation time courses in the presence of AITC remain reasonably monoexponential, they become multiexponential in the presence of menthol. The differential effects on relaxation rates are fully explained by assuming that AITC destabilizes the closed state, whereas menthol stabilizes the open state, with respect to the transition state. On the other hand, the mono- vs. multiexponential nature of the relaxation time courses are fully explained by assuming that the rates for binding/unbinding are rapid for AITC, but slow for menthol. These kinetic features allow the authors to define two classes (class I and II) of TRPM8 agonists.

Overall this is a beautiful study that promotes our understanding of the mechanisms of TRPM8 agonists. The experiments are well performed, and the analysis mostly adequate. The following requests aim to improve the clarity of the presentation.

Requested revisions:

1) It is stated several times that "AITC acts by destabilizing the closed channel, whereas menthol stabilizes the open channel" (Abstract; subsection “Defining Type I and Type II agonists”, first and second paragraphs; Discussion, second paragraph). It would be important to emphasize that these effects are understood by taking the transition state as a reference. E.g.: "AITC destabilizes the closed state, whereas menthol stabilizes the open state, relative to the transition state". This clarification is essential, because it is much more common to take the closed state as a reference, and both ligands act by stabilizing the open state relative to the closed state.

2) Along the same lines, we don't think the following argument is valid: "Type I ligands reduce the Gibbs free energy of the open state, which may facilitate obtaining stable channel protein in the open state for crystallography […]” In contrast, Type II ligands, which provoke a destabilization of the closed state, may be less prone to induce the formation of stable open channel-ligand complexes." Both types of ligands stabilize the open state relative to the closed state, i.e., they both increase the fraction of time the protein spends in the open state and so should similarly affect crystallography studies.

3) The finding that AITC binding selectively affects opening, while menthol binding selectively affects closing rate is interesting, and could be interpreted in terms of different *φ* values for the two binding sites. The *φ* value of a protein position reflects the relative timing of its movement upon pore opening, with large values reflecting early, small values late movement (Auerbach, 2007, J. Gen. Physiol. 13:543-546). The fact that AITC affects ΔG(T-C) but not ΔG(T-O) suggests that the AITC binding site has already reached its open-like conformation in the transition state, i.e., this region moves early during the opening conformational change (*φ* is large). In contrast, the fact that menthol affects ΔG(T-O) but not ΔG(T-C) suggests that the menthol binding site is still in its closed-like conformation in the transition state, i.e., this region moves late during opening (*φ* is small). By the way, relatively distant binding sites are also supported by the finding that the effects of simultaneously applied AITC and menthol are well described assuming independent binding and energetically additive effects.

4) The presentation of the modeling/fitting could be improved.

4.1) Using the Eyring prefactor *k_B_T/h* to compute ΔG^α†^ and ΔG^β†^ from the rates α_0_ and β_0_ is not adequate: it is generally believed that the prefactor for protein conformational transitions is much smaller than *k_B_T/h* (Auerbach, 2005; PNAS 102:87-92). In fact, computing the absolute values of these barrier heights is also not necessary: ΔΔG^α†^ and ΔΔG^β†^ values would suffice.

4.2) Table I should provide the full set of kinetic parameters, not (or not only) thermodynamic parameters: ΔΔG values could be expressed as fold changes in K_eq_, and α_0_ and β_0_ should be given instead of ΔG^α†^ and ΔG^β†^. This would make it easier for the reader to follow the discussion on the models shown in Figure 5—figure supplement 1 (or maybe print the rates onto those models?). Also, please specify how the rates for binding/unbinding in the closed state were handled during fitting: was k_on,c_ assumed identical to k_on,o_ with k_off,c_ constrained by detailed balance, or was the ratio k_on,c_/k_off,c_ constrained but the absolute values for both left free? In the latter case, please provide the fitted values in Table I.

4.3) For the global fitting please provide a concise summary with (i) a listing of all the parameters in the fit, (ii) the numbers and identities of fixed vs. free parameters, and (iii) the typical number of current traces fitted simultaneously.

5) The meaning of the slow and fast time constants plotted in Figure 4 are unclear to us. The symbols we expect represent fit parameters obtained by fitting a double-exponential function to the current time courses. But we don't understand the statement (Figure 4 legend): "Solid lines in (E) and (F) represent model predictions". The model predicts – for both types of ligand – 9 exponential components for each current relaxation. Which of those 9 time constants are plotted? The ones with the largest fractional amplitudes? But those fractional amplitudes will depend on ligand concentration, so different components will become dominant at different ligand concentrations. All in all, it might be better to plot a single empirical parameter here (as in E), instead of two arbitrary time constants. E.g., the time constant of the best monoexponential fit (even if that fit is bad)? Or the half-time of the relaxations?

6) The differential calcium entry in sensory neurons and HEK cells shown in Figure 7 is framed as arising from the ability of sensory neurons to fire action potentials. It would be cool to actually show this by blocking Na_v_ channels with TTx and showing that they then behave like HEK cells. We leave it to the discretion of the authors because it would require a new experiment.

---

## [Author Response]

*Requested revisions:*

*1) It is stated several times that "AITC acts by destabilizing the closed channel, whereas menthol stabilizes the open channel" (Abstract; subsection “Defining Type I and Type II agonists”, first and second paragraphs; Discussion, second paragraph). It would be important to emphasize that these effects are understood by taking the transition state as a reference. E.g.: "AITC destabilizes the closed state, whereas menthol stabilizes the open state, relative to the transition state". This clarification is essential, because it is much more common to take the closed state as a reference, and both ligands act by stabilizing the open state relative to the closed state.*

We fully agree that this was not correctly described, and have now clearly indicated that these effects are relative to the transition state.

*2) Along the same lines, we don't think the following argument is valid: "Type I ligands reduce the Gibbs free energy of the open state, which may facilitate obtaining stable channel protein in the open state for crystallography […]” In contrast, Type II ligands, which provoke a destabilization of the closed state, may be less prone to induce the formation of stable open channel-ligand complexes." Both types of ligands stabilize the open state relative to the closed state, i.e., they both increase the fraction of time the protein spends in the open state and so should similarly affect crystallography studies.*

This argument has been removed, and replaced by a discussion along the lines suggested below.

*3) The finding that AITC binding selectively affects opening, while menthol binding selectively affects closing rate is interesting, and could be interpreted in terms of different φ values for the two binding sites. The φ value of a protein position reflects the relative timing of its movement upon pore opening, with large values reflecting early, small values late movement (Auerbach, 2007, J. Gen. Physiol. 13:543-546). The fact that AITC affects ΔG(T-C) but not ΔG(T-O) suggests that the AITC binding site has already reached its open-like conformation in the transition state, i.e., this region moves early during the opening conformational change (φ is large). In contrast, the fact that menthol affects ΔG(T-O) but not ΔG(T-C) suggests that the menthol binding site is still in its closed-like conformation in the transition state, i.e., this region moves late during opening (φ is small). By the way, relatively distant binding sites are also supported by the finding that the effects of simultaneously applied AITC and menthol are well described assuming independent binding and energetically additive effects.*

This is an excellent suggestion. We have now included a discussion referring to the *φ*-value as described in the work of Auerbach and colleagues, and the potential implications for the two binding sites.

*4) The presentation of the modeling/fitting could be improved.*

*4.1) Using the Eyring prefactor k_B_T/h to compute ΔG*^α†^*and ΔG*^β†^*from the rates α*_0_
*and β*_0_
*is not adequate: it is generally believed that the prefactor for protein conformational transitions is much smaller than k_B_T/h (Auerbach, 2005; PNAS 102:87-92). In fact, computing the absolute values of these barrier heights is also not necessary: ΔΔG*^α†^*and ΔΔG*^β†^
*values would suffice.*

*4.2) Table I should provide the full set of kinetic parameters, not (or not only) thermodynamic parameters: ΔΔG values could be expressed as fold changes in K_eq_, and α*_0_
*and β*_0_
*should be given instead of ΔG*^α†^
*and ΔG*^β†^*. This would make it easier for the reader to follow the discussion on the models shown in Figure 5—figure supplement 1 (or maybe print the rates onto those models?). Also, please specify how the rates for binding/unbinding in the closed state were handled during fitting: was k_on,c_ assumed identical to k_on,o_ with k_off,c_ constrained by detailed balance, or was the ratio k_on,c_/k_off,c_ constrained but the absolute values for both left free? In the latter case, please provide the fitted values in Table I.*

*4.3) For the global fitting please provide a concise summary with (i) a listing of all the parameters in the fit, (ii) the numbers and identities of fixed vs. free parameters, and (iii) the typical number of current traces fitted simultaneously.*

We apologize that the explanation and presentation of the fitting was not optimal, and agree that the ΔG values are not really adequate, since the absolute value of the prefactor is unknown. We now include the undetermined transmission coefficient (κ) to the equations to indicate this. Moreover, in Table 1 we now replaced the ΔG values by actual forward and backward rates of the unliganded channel at 0 mV. More details on the fitting, including the numbers and identities of the free and fixed parameters are now provided in the Methods and in Table 1.

*5) The meaning of the slow and fast time constants plotted in Figure 4 are unclear to us. The symbols we expect represent fit parameters obtained by fitting a double-exponential function to the current time courses. But we don't understand the statement (Figure 4 legend): "Solid lines in (E) and (F) represent model predictions". The model predicts – for both types of ligand – 9 exponential components for each current relaxation. Which of those 9 time constants are plotted? The ones with the largest fractional amplitudes? But those fractional amplitudes will depend on ligand concentration, so different components will become dominant at different ligand concentrations. All in all, it might be better to plot a single empirical parameter here (as in E), instead of two arbitrary time constants. E.g., the time constant of the best monoexponential fit (even if that fit is bad)? Or the half-time of the relaxations?*

We apologize for not better explaining this figure. What we found is that, in the presence of menthol, both the experimental data and the modelled currents were excellently fitted by two exponentials. That the modeled currents are “pseudo-bi-exponential”, despite the fact that the model theoretically predicts 9 exponentials, can be attributed to (1) fractional amplitudes of several time constants being negligibly small, and (2) clustering of time constants around two main values. Thus, to obtain the solid lines in Figure 4, current traces during voltage steps as in Figure 4 were modeled for different menthol concentrations using the mean parameters of Table 1, and then fitted with a double-exponential function to obtain the fast and slow time constants. Fitting mono-exponential functions to the modeled (and experimental data) in the presence of menthol provides indeed relatively “bad” fits, but nevertheless provides a good concurrence between experimental and model data. This plot is now provided in Figure 4—figure supplement 1, along with fits & residuals of experimental and modeled data using mono- and bi-exponential functions.

6) The differential calcium entry in sensory neurons and HEK cells shown in Figure 7 is framed as arising from the ability of sensory neurons to fire action potentials. It would be cool to actually show this by blocking Na_v_ channels with TTx and showing that they then behave like HEK cells. We leave it to the discretion of the authors because it would require a new experiment.

We thank the referees for this excellent suggestion, and now included new data in the presence of TTX. We found that TTX indeed reduces the responses to menthol relative to AITC, similar to the behavior in HEK cells. This is now presented in Figure 7.